# Dynamic Response Analysis of Buried HDPE Pipes under Vibration Compaction Considering the Influence of Buried Depth and Filling Modulus

**Chongqian Ma, Xuan Wang \*, Jiasheng Zhang, Hao Luo and Yu Zhang**

Department of Civil Engineering, Central South University, Changsha 410075, China
* Correspondence: dddebug@csu.edu.cn

**Abstract:** During pavement construction, the gravity load and vibration excitation from vibratory rollers can seriously affect the safety of underground pipelines. However, research on the dynamic response of buried pipelines under the action of large vibration rollers is rarely reported. Therefore, the dynamic response of a high-density polyethylene (HDPE) double-wall corrugated pipe under a vibratory roller was studied via field testing and a three-dimensional nonlinear finite-element model was established. This model was used to analyze the influence of filling property and vibration frequency on the dynamic response of an HDPE pipe under a vibration load from a vibratory roller. The results reveal that with the increase of compaction times, the backfill soil keeps changing between over-compaction (loose) and compaction states, and the pipe top pressure also keeps changing. Moreover, at a shallow burial, the pipe top pressure is more obvious when the compaction degree changes. The elastic modulus of filling soil within a certain range can effectively reduce the stress deformation of the pipeline under vibration compaction. However, when the elastic modulus of filling soil exceeds 10 times the initial elastic modulus, the deformation of the pipeline becomes greater than the initial value.

**Keywords:** HDPE pipe; vibratory compaction; pipeline strain and stress; pile-soil interaction

## 1. Introduction

During pavement construction, the vibration load of vibratory rollers has an obvious influence on buried pipelines. The additional vibration load generated by rollers poses a serious threat to the overall performance of pipelines, thereby shortening the service cycle of buried pipelines. However, there are few reports on the variation of the dynamic response of buried pipelines under the action of large vibratory rollers. The buried depth and backfill properties are key factors affecting the mechanical response of pipelines [1]. Therefore, studying the mechanical response rules of buried pipelines from vibration compaction during construction and the influence of buried depth and filling properties has important significance for guiding the evaluation of pipeline safety.

There has been considerable previous research on the dynamic response of underground pipelines under earthquake action [2–10], including the calculation and analysis method of the dynamic response of pipelines during an earthquake, the mechanisms of pipe–soil interactions, the influence of various factors on the dynamic response of pipelines, and the calculation method for the seismic stress between pipelines. These studies provide some ideas for the study of the influence of vibratory rollers on buried pipelines. For example, Moleskin et al. [9,10] proposed a thin circular cylindrical shell model for the earthquake behavior of buried pipes, which took into account the bucking and fracture modes of the buried pipelines. This model provides an idea for the three-dimensional force analysis of a pipeline under dynamic load. However, compared with seismic load, the vibration loads of rollers are quite different in vibration frequency, scope, and time of

action. Therefore, the previous research methods cannot be used to fully study the dynamic response of underground pipelines caused by vibratory rollers.

At present, there are few reports about the dynamic response of buried pipelines under the action of a large vibratory roller. Abolmaali et al. [11] studied the stress-strain characteristics of cement pipelines under the action of a vibratory roller, predicted the initiation and propagation of cracks, and studied the influence of the compaction degree of pipe-side soil on pipeline deformation. However, in this paper, compaction forces were applied only at four specific locations without considering the impact of roller speed. Alzabeebee et al. [12] studied the behavior of buried pipes under static and moving traffic loads using a robust finite element analysis. The results show that the maximum pipe crown displacement is greatly affected by the load velocity, and the increasing load speed causes a small decrease in the induced pipe displacement. Therefore, if the load movement is not considered, the results are conservative and inconsistent with the actual situation. In addition, Abolmaali's research did not involve flexible pipes.

The influence of soil cover depth on pipes has been studied extensively. Lohnes et al. [13] developed a numerical model and recommended a minimum soil cover of 305 mm, which is identical to the one proposed by Watking and Kang [14,15]. Rakitin et al.'s [16] experiments on concrete pipelines show that when the buried depth of the pipeline increases, the increase of the bending moment of the pipeline caused by the self-weight of the soil is much smaller than the decrease of the bending moment caused by the traffic load. Saboya et al. [17] studied the deformation of the steel pipe buried in cohesionless soil under moving surface loads. Experiments showed that when the pipe was installed in a shallow depth, the cross-section of the pipe changed from a vertical ellipse to a horizontal ellipse under a moving surface load. Tian et al. [18] studied the influence of buried depth on the impact force and acceleration of pipelines under impact load. The results showed that the maximum acceleration and impact force on the pipeline decreased with the increase of buried depth because the vibration wave attenuated rapidly with the increase of distance. Tafreshi and Khalaj [19] studied the mechanical response of flexible pipes under cyclic loading through a laboratory model test. The results showed that when the buried depth of the pipeline was greater than 2.5D, the deformation of the pipeline was almost affected by the buried depth. Elshesheny [20] also reached the same conclusion. The above studies mainly focused on the impact loads and traffic loads. Compared with these two loads, the load and loading frequency of the roller are larger. At the same time, the object of the roller is uncompacted soil, and the degree of compaction of the soil will change dramatically when the roller load is applied. Therefore, the influence of the change of compactness on the pipeline must be considered.

The soil around the pipeline plays a role in transmitting load and supporting the pipe [21]. Therefore, the parameters of the soil are also important factors influencing the mechanical response of the pipeline. Neay et al. pointed out that the ultimate strength of the pipe-soil system increased linearly with increasing the backfill degree of compaction [13]. However, the well-compacted soil makes larger deformations of the pipeline due to compaction [22]. Under dynamic loads, on the one hand, the soil around the pipe should provide sufficient support for the pipeline, which requires sufficient stiffness of the soil. On the other hand, the soil should have sufficient damping to reduce the effect of dynamic load on the pipeline. However, the compliance of soil required for damping the dynamic loads is opposed to the stiffness of the material required for the task of load distribution and carrying the load [23].

At present, the research on the dynamic response of buried pipelines mainly focuses on seismic loads, traffic loads, and impact loads, while loads of vibratory rollers are rarely studied. In this paper, the field test was carried out to study the stress and deformation lows of HDPE pipe under the action of a vibratory roller. Additionally, the infinite element boundary and shell element models were used to establish a three-dimensional nonlinear finite-element model. This model is used to analyze the influence of the filling property and vibration frequency on the dynamic response of HDPE pipe under the vibration load

of a roller and the influence of buried depth and filling property on the pipeline under earth pressure and the vibration load of the roller is compared.

## 2. Field Testing

### 2.1. Test Materials

Double-walled corrugated pipe with a diameter of 800 mm was used in field testing. The specific parameters are shown in Table 1. The instruments used for data measurements and analysis include strain gauges, dynamic earth pressure boxes, a vibration roller, and an acquisition device. The mass of the vibratory roller was 22 t, the vibration frequency was 28 Hz/33 Hz, and the vibration amplitude was 1.86 mm/0.93 mm. The specific parameters are shown in Table 2.

**Table 1.** Basic parameters of HDPE pipe used in testing.

| Type of Pipeline | Inside Nominal Diameter | Elastic Modulus | Poisson's Ratio | Circumferential Stiffness | Density |
|---|---|---|---|---|---|
| Double-wall corrugated pipe | 800 mm | 800 MPa | 0.4 | 10 kN/m$^2$ | 950 kg/m$^3$ |

**Table 2.** Basic parameters of vibratory roller used in testing.

| Mode of Vibration | Vibration Frequency | Vibration Amplitude | Excitation Force Amplitude | Static Pressure of the Roller |
|---|---|---|---|---|
| Weak vibration mode | 33 Hz | 0.93 mm | 290 kN | 110 kN |
| Strong vibration mode | 28 Hz | 1.86 mm | 374 kN | |

### 2.2. Instrument Layout and Test Plan

A trapezoidal ditch was excavated for pipeline burial in the original subgrade with a slope of 1:0.33. The compactness of backfill soil followed regulatory standards [24], as shown in Figure 1. The circumference of the pipe was filled with stone chips and stone powder, and 50 cm of stone powder was used as fill above the pipe, followed by graded gravel and soil-rock mixture. The thickness of each layer was 30 to 40 cm after compaction. The design elevation of the pavement was 46.02 m, the design elevation of the pipe bottom was 42.71 m, and the buried depth of the pipe was 3.31 m, as shown in Figure 2.

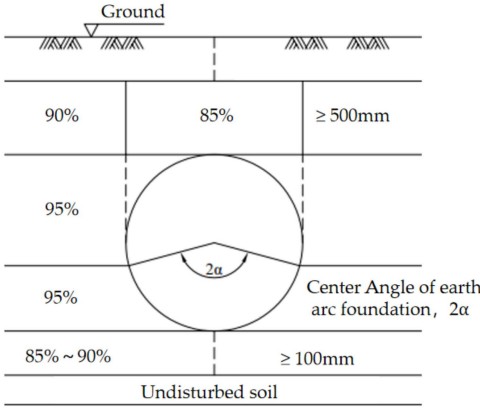

**Figure 1.** Trench soil backfill requirements.

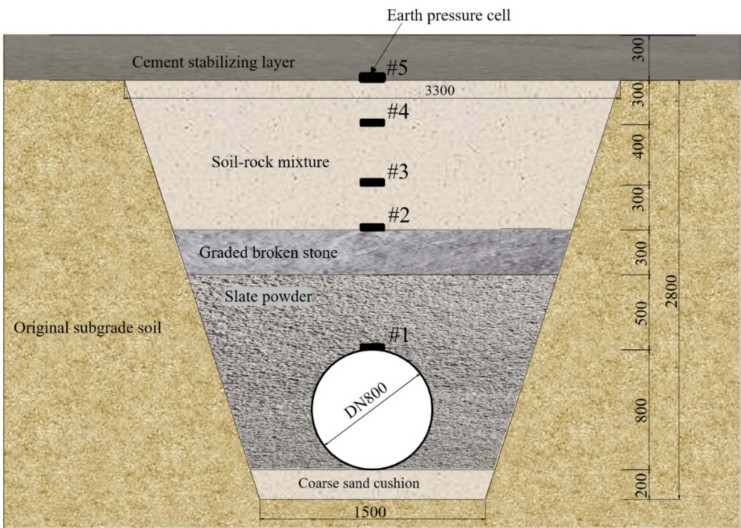

**Figure 2.** Layout of earth pressure cells in the filling soil.

Four strain gauges were arranged at the top and sidewall of the pipe. The strain gauge arrangement is shown in Figure 3. A total of five earth pressure cells were arranged. The #1 cell was arranged at the top of the pipeline, and the #2 earth pressure cell was placed 80 cm above the #1 cell. The #3 cell was placed in the soil-rock mixture layer, 30 cm away from the #2 cell. The #4 cell was placed at 40 cm above #3, and the #5 cell was installed after laying two layers of an earth-rock mixture, 30 cm away from the #4 cell. The detailed layout is shown in Figure 2.

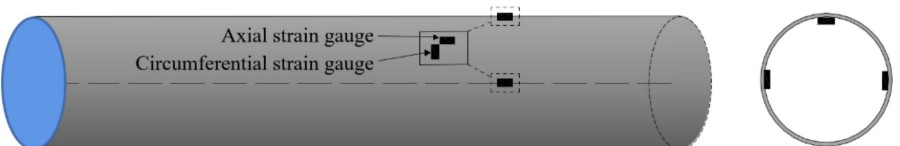

**Figure 3.** Layout of pipeline strain measuring points.

To analyze the mechanical properties of buried pipelines under different buried depths and dynamic load frequencies, vibration compaction tests with different frequencies were carried out in three stages of backfill, which are shown in Table 3. The vibration frequency under the strong vibration mode was 33 Hz, and under the weak vibration mode was 28 Hz. Taking Condition 1 as an example, the roller advanced at a speed of 2.63 km/s from 15 m away from the measuring point in the weak vibration mode and stopped advancing and vibrating 3 m away from the measuring point. When the signal was stable, the roller reversed and returned to the starting point. To fully compact the filler, the process was repeated ten times. The soil dynamic stress amplitude and pipeline dynamic strain amplitude represent the most dangerous state of the pipeline, and, thus, the test focused on monitoring the dynamic pressure and pipeline strain amplitude under different test conditions.

**Table 3.** Field vibration compaction test conditions.

| Working Condition | Test Content | Mode of Vibration | Compaction Times | Buried Depth of Pipeline |
|---|---|---|---|---|
| Condition 1 | Pipeline strain, earth pressure | Weak vibration mode | 10 | 1.5 m |
| Condition 2 | Pipeline strain, earth pressure, acceleration of the vibratory roller | Strong vibration mode | 12 | 1.8 m |
| Condition 3 | Pipeline strain, earth pressure | Strong vibration mode | 10 | 2.1 m |

*2.3. Analysis of Soil Pressure Amplitude on the Pipe Top and Dynamic Strain Amplitude of the Pipeline*

The variation of dynamic earth pressure amplitude at the pipe top at various compaction times is shown in Figure 4. The figure shows that with increasing filling depth, the amplitude of earth pressure on the pipe roof decreased, indicating that the influence of dynamic load on the pipeline decreases with an increase in the buried depth of the pipeline. After the ninth compaction, the soil pressure on the pipe roof in Condition 3 was close to that in Condition 2, and the increase of the buried depth of the pipeline had little contribution to the decrease of the top pressure of the pipeline. Therefore, a 1.8 m burial depth can be taken as the optimal depth for the project.

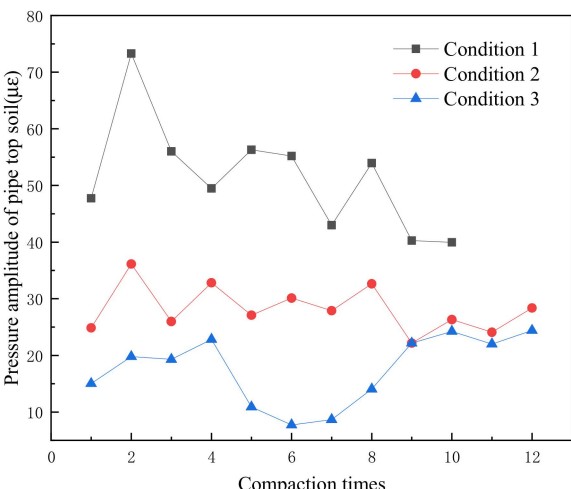

**Figure 4.** Amplitudes of pipe roof soil pressure.

The amplitude of dynamic earth pressure on the pipe top is affected by compaction times. Meanwhile, the vertical acceleration of the vibratory roller in Condition 2 presented the same trend as the pressure on the pipe top, and its amplitude fluctuated with the compaction times, as shown in Figure 5. At the first compaction, the filling material was relatively loose, and the vibration energy of the vibratory roller was mainly dissipated in the form of plastic deformation energy storage of the filling material. At the second compaction, the filling material became dense, and the plastic energy storage was reduced, so the acceleration amplitude showed a rebound phenomenon. However, at the third compaction, the soil was over-compacted, and the filling material was re-compacted to absorb energy, so the acceleration amplitude decreased. The same trend between acceleration and pipe top pressure indicates that the pipe top pressure increases with the increase of backfill compaction degree. It is noted that at the ninth compaction, the vibration acceleration amplitude reached the maximum value of 5.178 g, indicating that the filling material had reached another better compaction state. The minimum vibration acceleration amplitude measured at the tenth compaction also confirmed this. Due to the large distance between the pipe top and the ground, in Condition 3, the pipe top soil pressure showed the same change trend between several compactions and was not as sensitive to the compaction

times as the other two conditions. The maximum and minimum values of the pipe top pressure amplitude in Condition 1 were 34.56 kPa, which was much larger than that in Condition 2 and Condition 3. This indicates that when the pipe is shallowly buried, the pipe top pressure is significantly affected by the compaction degree. The maximum and minimum values of the amplitude of the pipe top pressure in Condition 1 differed by 34.56 kPa, which was much larger than that in Condition 2 and Condition 3. This indicates that when the pipeline is shallowly buried, the pipe top pressure is significantly affected by the compaction.

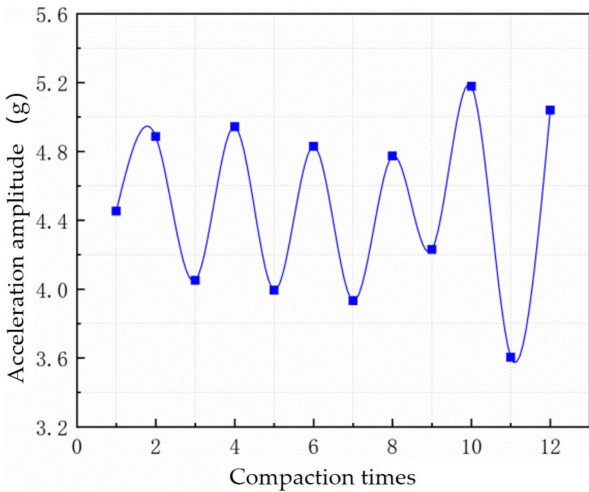

**Figure 5.** The vertical acceleration of the vibrating wheel in Condition 2.

Figure 6 shows the amplitude values of soil pressure at different burial depths under Condition 2. It can be seen from the figure that the variation trend of the amplitude value of the #4 soil pressure cell closest to the ground is opposite to that of the other three soil pressure boxes, which is due to the fact that the surface filling soil is compacted before the bottom filling material.

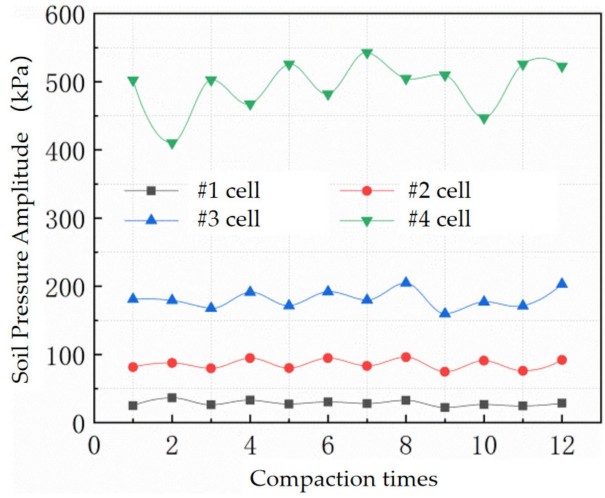

**Figure 6.** The amplitude values of soil pressure at different burial depths under Condition 2.

The axial and circumferential strain amplitudes along the pipe side and over the top are shown in Figures 7 and 8. It can be seen from the figure that the strain at the top of the pipe is mainly axial strain, while at the side of the pipe, circumferential strain is greater than axial strain. In addition, the strain values at the side of the pipe are all greater than those at the top of the pipe. This results from the vertical compression of the pipeline under the action of vertical pressure and the relative displacement occurring between the soil

above the pipeline and the soil at the pipeline side, which causes the soil arching effect. Therefore, part of the pressure above the pipeline was borne by the soil along the pipe side, the soil pressure thus increased, and the circumferential strain value at the pipeline side was greater than that of other parts. The strain above the pipe top was mainly axial. At the same time, the vertical compression deformation of the pipeline caused the pipeline to expand laterally, and the soil on the side of the pipe restricted this deformation, thus exerting a great force on the pipe. Therefore, the strain on the side of the pipe was greater than that of other parts.

Due to the above reasons, the force on the pipe side was particularly complex, so its trend of change with the number of compactions was slightly different from the pipe top pressure. The trend of hoop strain of the pipe top was consistent with the pipe top pressure. For the axial strain of the pipe top, because the compactor traveled in the direction of the length of the pipe, the axial strain was greatly affected by the adjacent parts, so its trend of change was also different from the pressure of the pipeline.

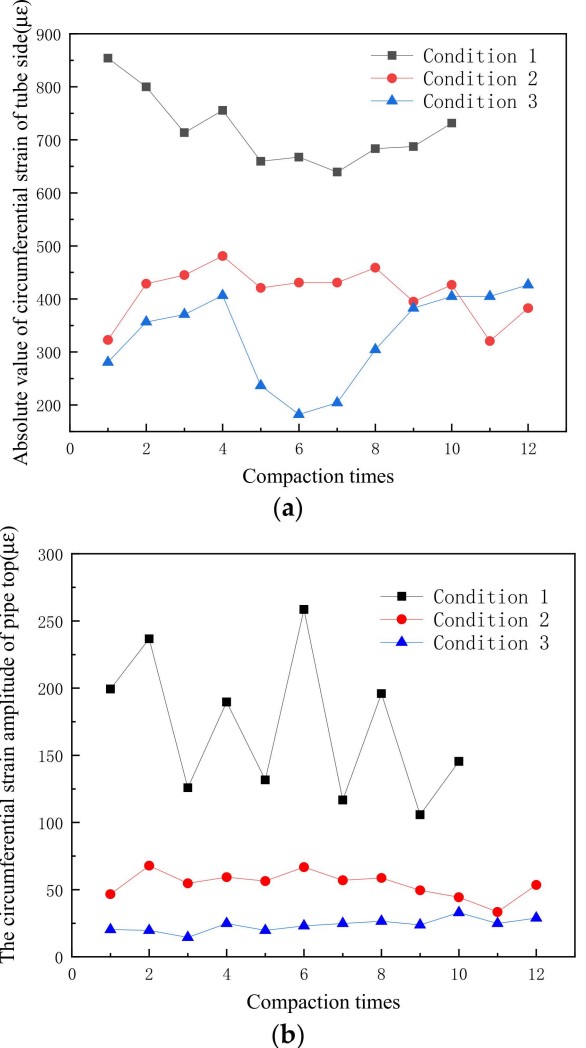

**Figure 7.** Absolute values of the circumferential strain of (**a**) pipe side and (**b**) pipe roof.

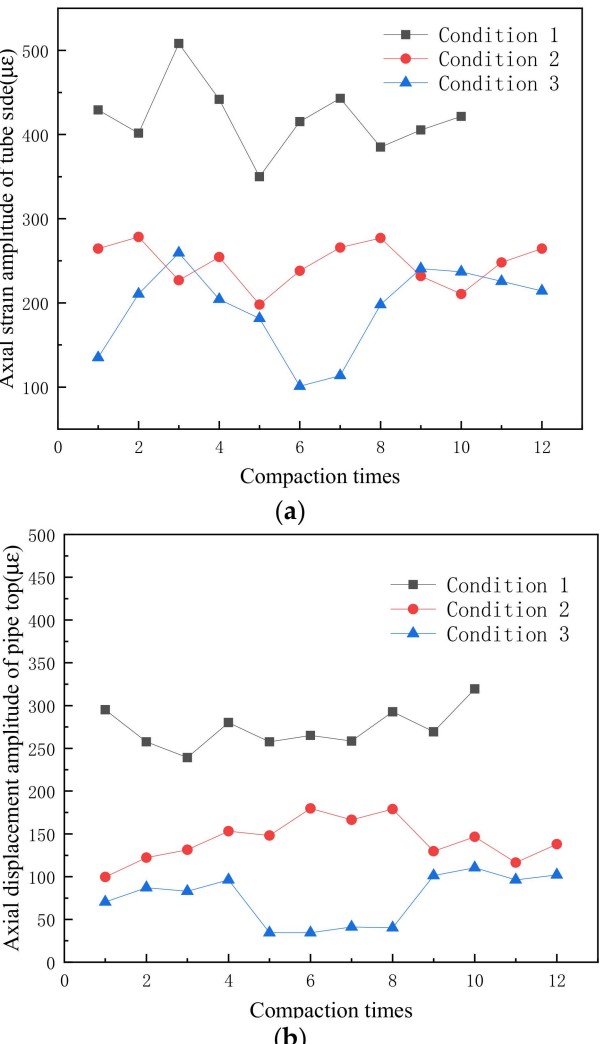

**Figure 8.** Absolute values of the axial strain of (**a**) pipe side and (**b**) pipe roof.

### 2.4. Comparison of Measured and Calculated Earth Pressure

At present, the common methods for calculating the dynamic earth pressure over the buried pipeline are the G Eason formula and the distributing angle method. Based on the elastic mechanics equation, G Eason [25] obtained the steady-state response of semi-infinite space under vertical point load using a three-dimensional Fourier transform and extended it to the steady-state response of semi-infinite space under uniform rectangular load. The stress response expression is as follows:

$$\sigma_x = -\frac{T}{\pi^2} \int_0^{\frac{\pi}{2}} \frac{\cot\theta}{H} \left\{ \left(1 - \frac{1}{2}\alpha_2^2 \cos^2\theta\right)\left(1 + \frac{1}{2}\alpha_2^2 - \alpha_1^2\right)v_1 - \gamma_1\gamma_2 v_2 \right\} d\theta \tag{1}$$

$$\sigma_y = -\frac{T}{\pi^2} \int_0^{\frac{\pi}{2}} \frac{1}{H\sin\theta\cos\theta} \left\{ \left(1 - \frac{1}{2}\alpha_2^2 \cos^2\theta\right)\left[\sin^2\theta + \left(\frac{1}{2}\alpha_2^2 - \alpha_1^2\right)\cos^2\theta\right]v_1 \\ - \gamma_1\gamma_2 v_2 \sin^2\theta \right\} d\theta \tag{2}$$

$$\sigma_z = \frac{T}{\pi^2} \int_0^{\frac{\pi}{2}} \frac{1}{H\sin\theta\cos\theta} \left\{ \left(1 - \frac{1}{2}\alpha_2^2 \cos^2\theta\right)^2 v_1 - \gamma_1\gamma_2 v_2 \right\} d\theta \tag{3}$$

$$\begin{cases} \alpha_2 = \beta\alpha_1 = \dfrac{V}{c_2} \\ H = \left(1 - \dfrac{1}{2}\alpha_2^2\cos^2\theta\right)^2 - \gamma_1\gamma_2 \\ u_{1,2} = \log\left[\dfrac{(a\cos\theta - b\sin\theta)^2 + \gamma_{1,2}^2 z^2}{(a\cos\theta + b\sin\theta)^2 + \gamma_{1,2}^2 z^2}\right] \end{cases} \tag{4}$$

where $c_2$ is the shear wave velocity in semi-infinite space.

The distributing angle method [26] assumes that the surface load on the pavement is transmitted downward along a specific diffusion angle and that the upper load is uniformly distributed in the isometric plane after transmission. According to the above assumptions, the vertical pressure generated from the excitation force generated by the vibratory roller vibration wheel at the top of the pipe can be obtained from:

$$q_{vκ} = \frac{k_d Q_{vκ}}{(L + 1.4H)(B + 1.4H)} \tag{5}$$

where $q_{vk}$ is the vertical pressure produced by the vibrating wheel on the pipe top (kN/m), $Q_{vk}$ is the contact force between the vibratory roller and the soil (kN), $L$ is the effective contact width between the vibrating wheel and the soil (m), $B$ is the length of the vibration wheel (m), $H$ is the pipe-jacking soil-filled height, and $k_d$ is the dynamic coefficient.

The earth pressure values obtained from the G Eason formula, the distribution angle method, and the field test during the ninth compaction are provided in Figure 9. It is observed from the figure that the variation law of the dynamic earth pressure amplitude in the subgrade with the depth calculated using the G Eason formula and the distribution angle method are consistent with the measured values. When the depth of the measuring point is less than 0.6 m, the dynamic earth pressure amplitude calculated by the G Eason formula is closer to the field measured value. In addition, it can be seen that the dynamic earth pressure amplitude at each depth measurement point calculated by the distribution angle method (when the buried depth is greater than 0.7 m) is the closest to the measured value. However, in the subgrade surface area, the distribution angle method underestimates the magnitude of the dynamic earth pressure produced by the vibratory roller. Therefore, in engineering, the G Eason formula can be used to predict the dynamic earth pressure amplitude above the pipe when the depth of the measuring point is less than 0.6 m. When the depth of the measuring point is greater than 0.7 m, the distribution angle method should be used to predict the dynamic earth pressure amplitude.

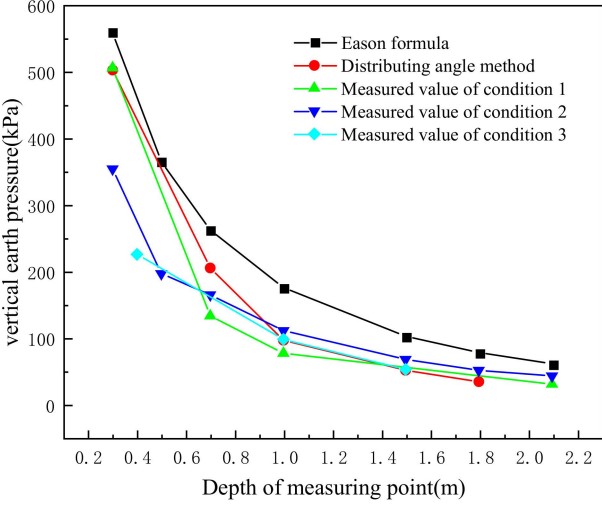

**Figure 9.** Earth pressure calculated by different methods.

## 3. Numerical Simulation

To obtain the circumferential and stress and strain distribution forms of the pipeline, a finite-element model was established, using the commercial finite element software Abaqus

6.14, according to the field conditions, and the rationality of the model was verified using the field test data. The model was also used to examine the influence of backfill properties and vibration frequency on the pipeline.

### 3.1. Establishment and Validation of the Numerical Model

In the model, the slope of the trapezoidal ditch was 1:0.33, the width of the bottom of the ditch was 1.5 m, the height to earth up was 2.1 m, the bottom of the ditch was made of a 200 mm thick cushion (A1), and the supporting angle of the foundation of the bottom was 120°. A2 was the backfill on the side of the pipe, A3 was the backfill on the top of the pipe, and A4 was the soil on both sides of A3. To prevent the reflection of vibration energy by fixed boundary in Abaqus dynamic analysis, which may cause errors, the model boundary adopted an infinite element boundary to simulate the far-field infinite soil [27,28]. The size of the model was set to 8 m × 9 m × 20 m, as shown in Figure 10. According to the study by George et al. [29], the soil outside the three-fold pipe diameter has little influence on the soil of the flexible pipe, so the size used in the model is reasonable. In order to improve the calculation accuracy, the backfill in the trench was divided into smaller elements. Before determining the element size, several comparisons were made. The results show that when the element size of backfill soil was less than 0.04 m × 0.04 m × 0.1 m, the simulation results were almost not affected by the element's size. Therefore, the maximum size of the backfill element was set to 0.03 m × 0.03 m × 0.07 m. The constitutive relation of soil adopted the classical Mohr-Coulomb constitutive. According to the compaction degree and physical properties of the soil at different positions, the material models such as Young's modulus, internal friction angle, and cohesion were obtained by referring to the relevant specifications [24,26], as shown in Table 4. The implicit method was used in the model.

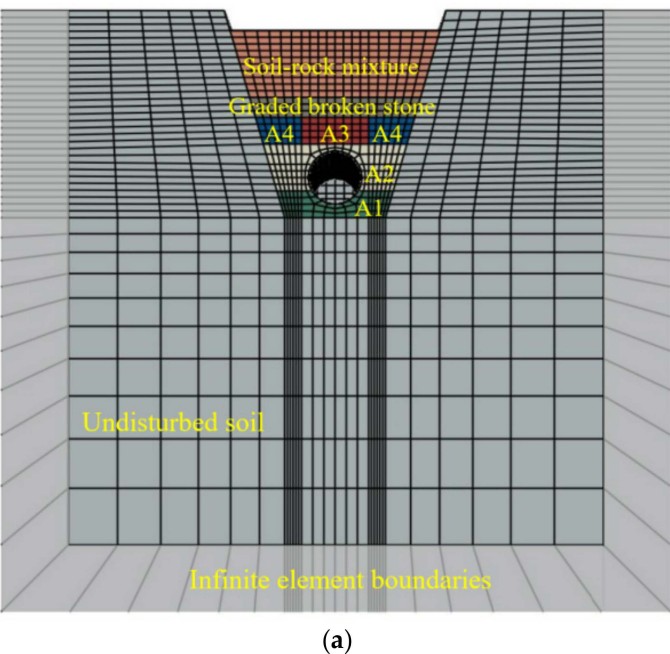

(**a**)

**Figure 10.** *Cont.*

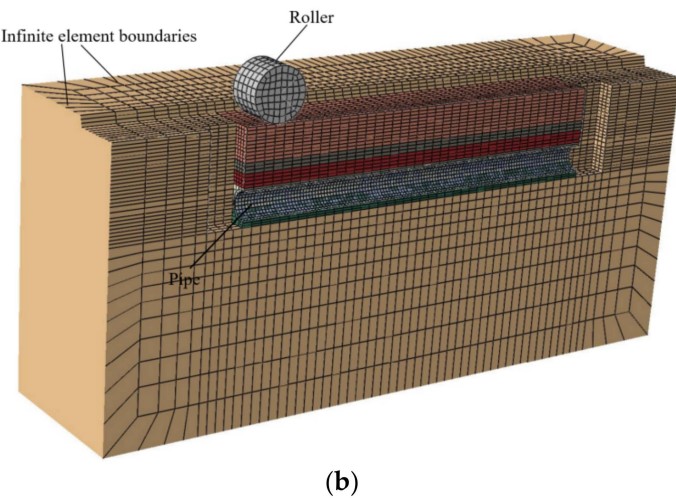

**(b)**

**Figure 10.** Finite element model: (**a**) cross-section of finite element model and (**b**) longitudinal section of finite element model.

**Table 4.** Physical parameters of backfill.

| Types of Backfill Soil | $\rho$ (kg/m³) | Modulus of Compression (MPa) | $\upsilon$ | Angle of Internal Friction (°) | Force of Cohesion (kPa) |
|---|---|---|---|---|---|
| Soil-rock mixture | 2100 | 72 | 0.17 | - | - |
| Graded broken stone | 2300 | 120 | 0.35 | 38 | 1 |
| A4 | 1600 | 7 | 0.3 | 35 | 15 |
| A3 | 1600 | 5 | 0.3 | 26 | 18 |
| A2 | 1600 | 7 | 0.25 | 26 | 16 |
| A1 | 1690 | 10 | 0.25 | 48 | 10 |
| Undisturbed soil | 2000 | 30 | 0.3 | 28 | 20 |

To improve calculation accuracy and efficiency, the HDPE double-walled corrugated pipe was converted into a three-dimensional straight wall pipeline with the same ring stiffness and outer diameter. The wall thickness of the straight wall pipeline can be calculated using Formula (6) [30]. The equivalent straight wall pipe model parameters are shown in Table 5. The constitutive model of the pipe adopted the classical Mohr-Coulomb constitutive.

$$PS = \frac{EI}{0.149r^3} \tag{6}$$

where $PS$ is the pipe stiffness, $E$ is the elastic modulus of pipe, $I$ is the inertia moment of the cross-section of the pipeline, and $r$ is the mean radius of the pipe.

**Table 5.** Numerical model parameters of equivalent straight wall pipeline.

| Double-Wall Corrugated Pipe | Modulus of Compression (MPa) | Poisson's Ratio | Circumferential Stiffness (kN/m²) | Inside Nominal Diameter (mm) | r (mm) | Equivalent Thickness (mm) |
|---|---|---|---|---|---|---|
| | 800 | 0.4 | 10 | 800 | 421.25 | 42.5 |

The vibration load of the vibratory roller is primarily generated by the front wheel vibration of the roller, and the rear vehicle as the static load had a negligible effect on the internal pressure of the subgrade. Therefore, in this numerical simulation, the vibratory roller was simplified into a wheel that can apply vibration load. The dynamic force, $F_{st}$, between the vibratory roller and the soil can be obtained by the two-degree-of-freedom

nonlinear simulation analysis of the vibratory roller [28]. $F_{st}$ is applied to the reference point of the vibratory roller as the compaction load of the vibratory roller. $F_{st}$ is expressed as:

$$F_{st} = \tfrac{1}{2}(F_s + G_a) + \tfrac{1}{2}(F_s - G_a)\sin(2\pi ft) \tag{7}$$

$$F_S = K_p(G_a + F_0) \tag{8}$$

where $F_0$ is the excitation force amplitude of the single-blade vibratory roller, $G_a$ is the static pressure of the roller, and $K_p$ is the additive coefficient considering the vibration effect. The value of $K_p$ can be obtained by regression analysis of the measured value of the reaction force during vibration compaction and the dynamic-to-static ratio. The formula for $K_p$ can be obtained by regression analysis:

$$K_p = 1.7 - 0.1\tfrac{F_0}{G_a} \tag{9}$$

According to the assumption of the two-degrees-of-freedom nonlinear simulation analysis of the vibratory roller [31], there is no slip between the roller and the ground, and no detachment occurs. Therefore, in the model, the tangential contact between the oscillating roller and the ground was defined as 'rough' contact. When the oscillating roller size was determined, its translational speed determined its rotational speed, so only the horizontal speed of the wheel was defined in the model. The parameters of the oscillating roller and $F_s$ are shown in Table 6.

**Table 6.** Model parameters of oscillating roller.

| Mode of Vibration | Ring Width | Radius | $G_a$ | $F_0$ | $F_s$ |
|---|---|---|---|---|---|
| Weak vibration mode | 2.13 m | 0.8 m | 110 kN | 290 kN | 574.55 kN |
| Strong vibration mode | | | | 374 kN | 658.24 kN |

The circumferential strain of the pipeline obtained via finite-element analysis and field tests is shown in Figure 11. The results of the finite-element analysis are in good agreement with the measured values, indicating that it is feasible to simulate the influence of vibratory roller construction on underground pipelines utilizing finite-element analysis. It also confirms that the loading method of the vibratory roller in the finite-element model in this study and the introduction of infinite element boundary to analyze the dynamic response of the buried pipeline under the action of the vibratory roller are reasonable.

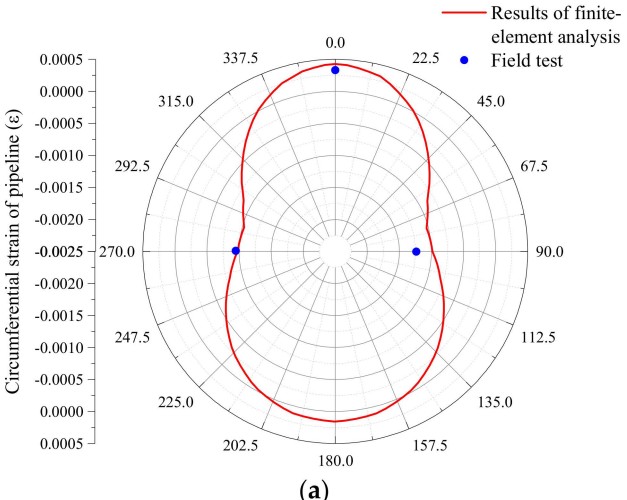

(**a**)

**Figure 11.** *Cont.*

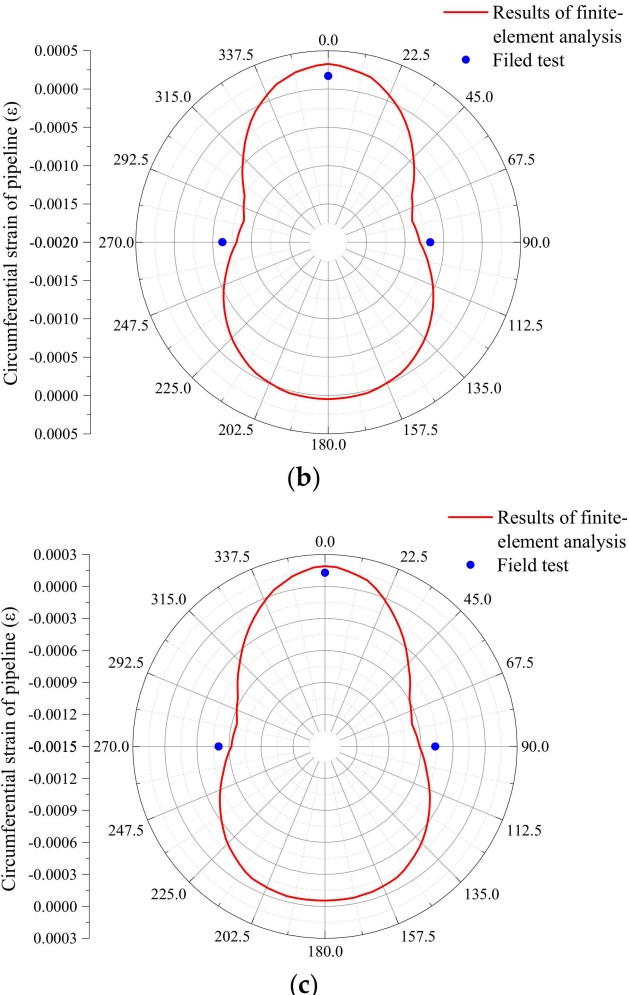

**Figure 11.** The circumferential strain of pipeline obtained by finite-element analysis and field tests: (**a**) Condition 1, (**b**) Condition 2, and (**c**) Condition 3.

*3.2. Influence of Burial Depth on Mechanical Response of Buried Pipeline*

(1)   Comparison of the soil arch rate and soil pressure calculated using the Chinese code with finite-element analysis.

The ratio of the earth pressure value to the self-weight stress of the soil at the measuring point is defined as the soil arch rate (VAF) [32]. The formula is as follows:

$$VAF = \frac{q}{\gamma H} \tag{10}$$

where $q$ is the earth pressure value at the top of the pipe, $\gamma$ is the unit weight of soil, and $H$ is the filling height at the top of the pipe.

Figure 12 provides the VAF calculated using the Chinese code method [24] and the finite-element analysis at the pipe top under soil pressure. The values of soil pressure on the pipe roof obtained by finite-element analysis and the Chinese code method are shown in Figure 13. It is observed from the figures that the VAF under each buried depth calculated by the finite-element model is less than 1, indicating that the soil arch effect occurs during backfill application during pipeline construction. In addition, with increasing buried depth, the VAF shows a decreasing trend, and the soil arch effect at the top of the pipeline is more obvious. The Chinese code [24] did not consider the influence of soil arching effect on the soil pressure on the pipe top, so the formula solution of the code was quite different from the finite-element analysis, and the difference between them further increased with

the increase of the buried depth of the pipe. When the buried depth reached 2.0 m, the calculated value of the standard formula was 1.58 times the finite element calculation value.

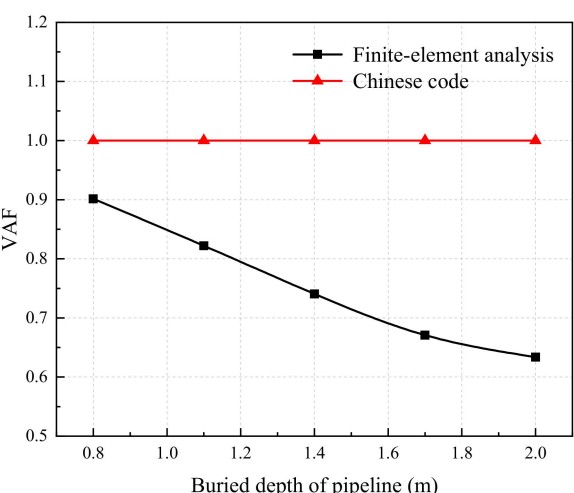

**Figure 12.** The VAF calculated using the Chinese code method and finite-element analysis.

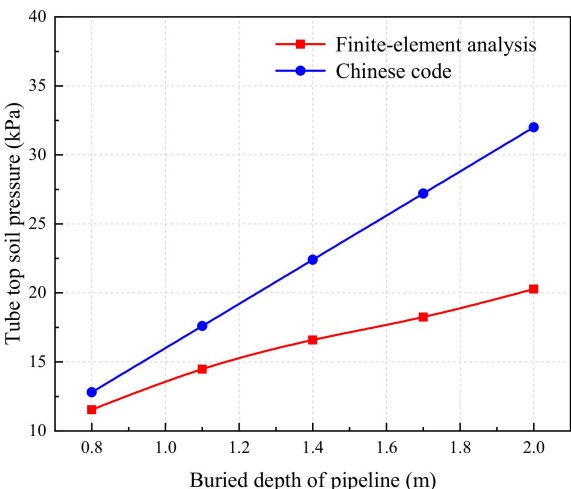

**Figure 13.** The soil pressure on the pipe roof.

(2)   Influence of buried depth on the mechanical response of pipeline

Figures 14 and 15 show the deformation of the pipeline and the displacement of the pipe top under strong and weak vibrations (vibration frequencies of loads are 28 Hz and 33 Hz, respectively). Figures 16 and 17 represent the deformation and displacement of the pipeline under earth pressure. It can be seen from these four figures that, under the earth pressure, the deeper the pipeline is buried, the greater the deformation of the pipeline and the displacement of the pipe top. The vertical deformation of the pipeline increased more rapidly than the horizontal deformation. When the buried depth increased from 0.8 m to 2 m, the lateral deformation increased by 1.02 mm, which was less than the 1.3 mm increase in vertical deformation. During backfilling, the compaction degree of the soil on the sides of the pipeline was higher than that on the top of the pipeline, which made the pipe less likely to deform to both sides. With increasing burial depth, the pipeline experiences horizontal deformation, which tends to compact the soil on the side of the pipeline. At the same time, the pressure from the top-filling soil will further compress the soil on the side of the pipeline. The compaction degree of the bottom-filling soil will also increase with the increasing pressure from the top-filling soil. Therefore, the increased rate of pipe top displacement decreases with the increase of buried depth.

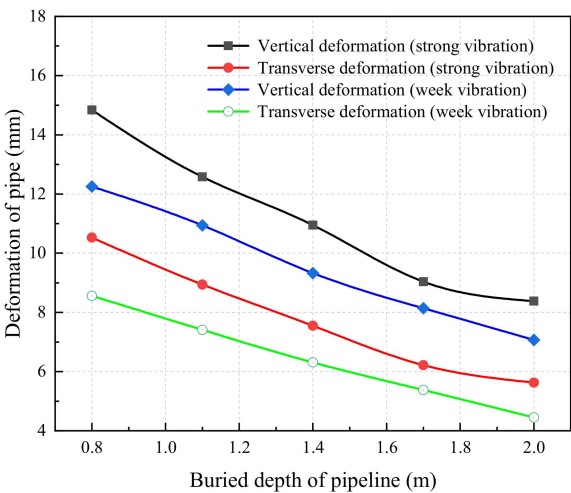

**Figure 14.** Deformation of the pipeline under different vibration modes.

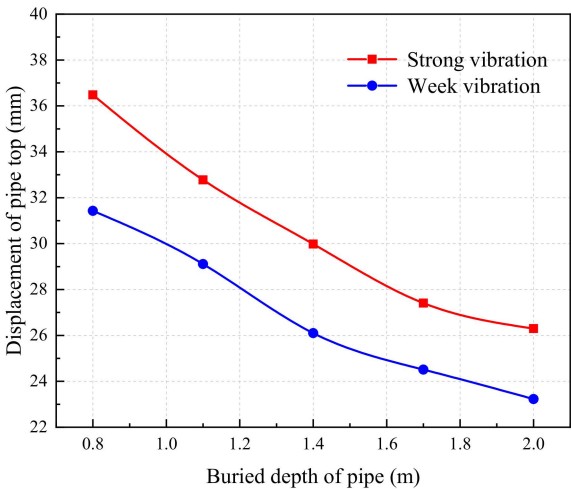

**Figure 15.** Displacement of the pipe top under different vibration modes.

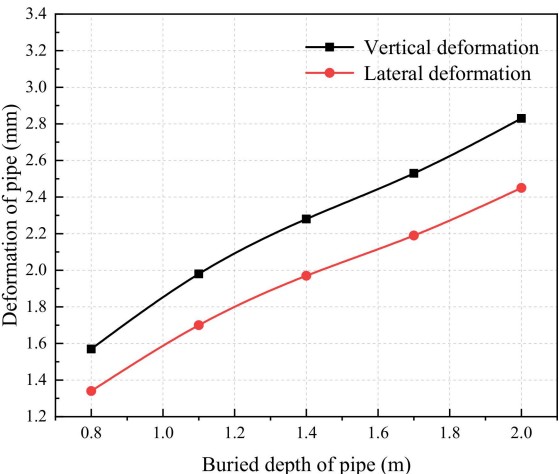

**Figure 16.** Deformation of the pipe under earth pressure.

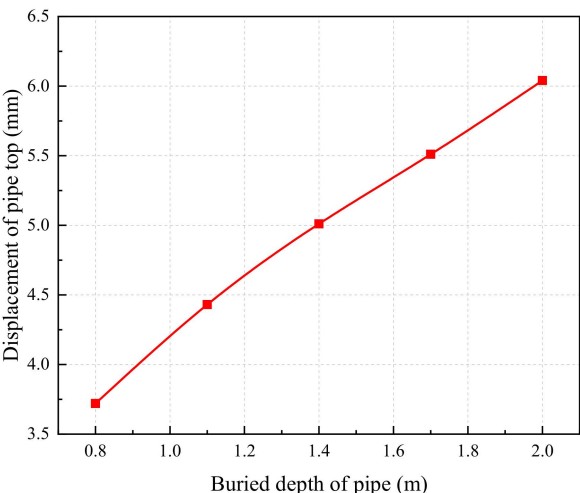

**Figure 17.** Displacement of the pipe top under earth pressure.

With an increase in filling height for both strong and weak vibration states, the deformation of the pipeline had a decreasing trend. This shows that with increasing buried depth, the speed of vibration load attenuation is greater than the growth rate of soil pressure. In addition, the vibration effect had a significant impact on the pipeline, making the deformation of the pipeline and the displacement of the pipe top significantly greater compared with the static effect. Additionally, it can be seen from Figures 14 and 15 that the strong vibration state had a greater impact on the pipeline than the weak vibration state. In the case of 0.8 m buried depth, the vertical deformation of the pipeline under strong vibration was 1.25 times that under weak vibration, and the displacement of the pipe top under strong vibration was 1.20 times that under weak vibration. When the buried depth was 1.7 m, the vertical deformation of the pipeline under weak vibration was 91% of that under strong vibration, and the displacement of the pipe top under weak vibration was 93% of that under strong vibration. This also demonstrates that the difference in the influence of weak and strong vibration states on the pipeline is inversely proportional to the increase in the buried depth.

According to the current regulation in China, vibratory rollers can only be used to compact the backfill soil after the burial depth of the pipe has reached 1.5 m, and the vertical deformation of the pipeline must be kept below 5% of the pipeline diameter (which is 40 mm in this study). The maximum vertical deformation of the pipeline at 0.8 m burial depth was 14.93 mm, which was much smaller than the regulation limit of 40 mm. From the graph, it can be seen that the displacement at the top of the pipeline was greater than the vertical deformation, indicating that the pipeline had experienced significant overall displacement. When the pipeline experienced settlement, a three-dimensional soil arching effect was formed above it, which could effectively resist some of the upper loads and reduce the impact of the upper loads on the buried pipeline [24]. Therefore, the deformation of the pipeline was much smaller than the regulation limit.

Figure 18 shows the Mises equivalent stress distribution of the pipe ring section under strong and weak vibration states, both of which are "butterfly-shaped" and symmetrical about the vertical direction. With the increase of burial depth, the equivalent stress of the ring section increases gradually, and the equivalent stress of the sidewall of the pipeline increases most obviously, followed by the top of the pipe and the slowest increase at the bottom of the pipe. Under the two vibration states, the maximum equivalent stress of the pipe ring section appears on both sides of the pipe, namely at 75° and 285°, which is similar to distribution under static force. Under the weak vibration state, the equivalent stress of the ring section at the top of the pipe is approximately the same as at the bottom of the pipe. While under the strong vibration state, the equivalent stress of the pipe top is significantly larger than the bottom.

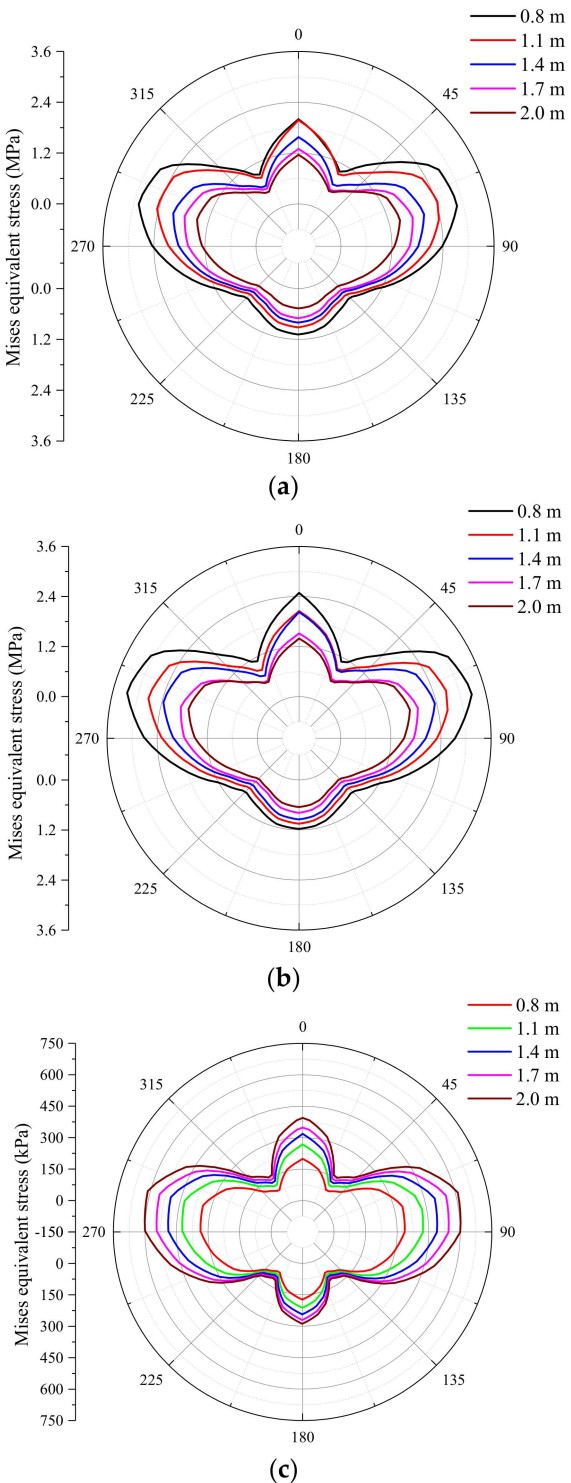

**Figure 18.** Equivalent stress of the pipeline under (**a**) weak vibration state, (**b**) strong vibration state, and (**c**) earth pressure.

### 3.3. Effect of Backfill Properties on Mechanical Response of Buried Pipeline

The numerical model was used for the numerical simulation of buried pipelines under the condition of 1.7 m buried depth. To control the influence of secondary factors on the analysis, the elastic modulus of the backfill at the top of the pipeline was changed to simulate different types of backfill, and the other parameters remained unchanged.

(1) Influence of backfill properties on pipeline mechanical response under earth pressure.

The displacement and soil pressure at the top of the pipeline change with the elastic modulus of backfill soil under the earth pressure, as shown in Figure 19. It can be seen that the displacement and earth pressure of the pipe top shows a downward trend with an increasing elastic modulus of the backfill soil. When the elastic modulus increases to 5 times the initial value, the displacement and the earth pressure of the pipe top decrease by about 10.0 and 15.3%, respectively. However, with a further increase in elastic modulus, the decreasing trend gradually diminishes. When the elastic modulus increases to 15 times the initial value, the pipe top displacement and the pipe top earth pressure decrease by 16.2 and 23.5%, respectively. According to the definition of soil arching rate, the increase of elastic modulus of the filling material causes the soil arching rate to decrease, demonstrating that the elastic modulus of the filling has a significant influence on the soil arching effect and that there is a positive correlation between them. The possible reason is that when the elastic modulus of backfill soil is large, its own resistance to deformation is strong, and the integrity is strong, so it can reduce the stress deformation of the pipeline under static force to a certain extent.

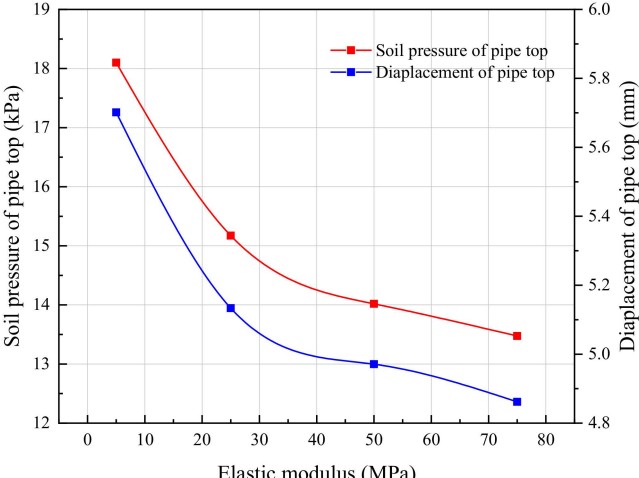

**Figure 19.** The displacement and soil pressure at the top of the pipeline.

Figure 20 shows the Mises equivalent stress and circumferential strain of the pipeline when backfilled with different properties. From Figure 20a, it can be seen that with the increase of elastic modulus of backfill, the equivalent stress on the middle ring section of the pipeline decreases gradually. Additionally, the maximum equivalent stress occurs at the centerline of the sidewall of the pipeline, and the equivalent stress at other positions does not decrease significantly with an increase in the elastic modulus of the fill. It can also be observed from Figure 20b that the variation law of circumferential strain is similar to that of equivalent stress, but the position where the maximum value of circumferential strain appears is slightly lower than that of equivalent stress. In summary, under the earth pressure, the increase of elastic modulus of backfill soil makes the soil arching effect more apparent, and more of the pipe top load is borne by the soil to the side of the pipe, which is also the reason for the decrease of pipe stress and strain. Therefore, the increase of elastic modulus of fill is conducive to the protection of pipelines.

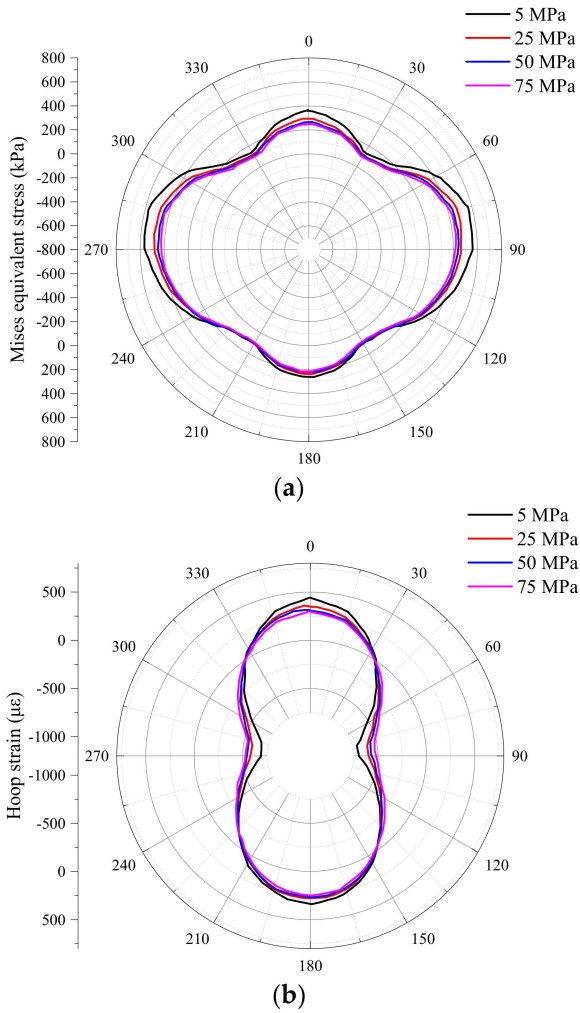

**Figure 20.** (**a**) mises equivalent stress and (**b**) hoop strain of pipeline.

(2)  Effect of backfill properties on the mechanical response of pipeline under vibration.

The mechanical response of the pipeline under the strong vibration mode is simulated by applying a vibration load on the filling surface. The results are shown in Figures 21 and 22. Compared with the static backfill state, the displacement amplitude at the top of the pipeline, soil pressure amplitude, and circumferential strain under the action of vibration compaction show different rules. With an increasing elastic modulus of the backfill, the displacement amplitude first decreases and then increases. When the elastic modulus of the filled soil increases to 25 MPa, the displacement amplitude of the pipe top and the soil pressure reach the minimum and then increase rapidly with the increase of the elastic modulus. When the elastic modulus of backfill soil is small, its integrity is poor, and it cannot effectively resist the action of vibration pressure. When the elastic modulus increases to a certain degree, its integrity is greatly improved, the resistance effect is obvious, and it can better protect the pipeline. However, the increase of elastic modulus will reduce the attenuation of vibration load in the soil. Therefore, when the elastic modulus exceeds a certain value, the improvement of integrity will be weaker than the influence of the decrease of vibration load attenuation speed on the pipeline. Therefore, under the action of vibration compaction and when the elastic modulus of filling is within a certain range, the stress deformation of the pipeline can be effectively reduced. However, when the elastic modulus of the filling exceeds 10 times the initial elastic modulus, the deformation of the pipeline is greater than the initial value.

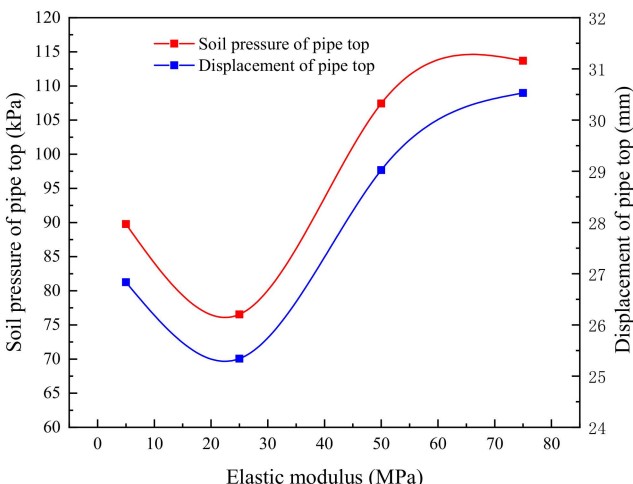

**Figure 21.** The displacement and soil pressure at the top of the pipeline under the strong vibration mode.

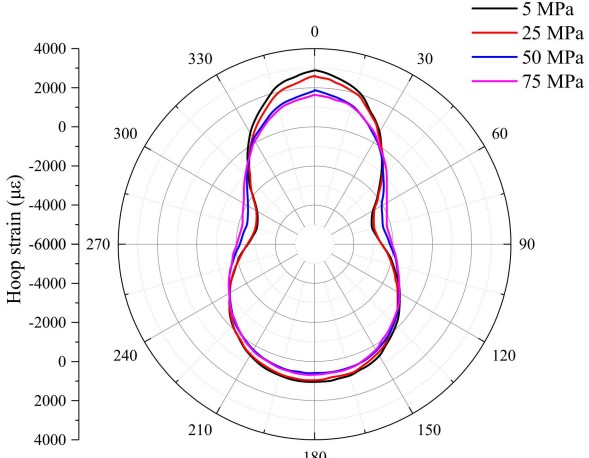

**Figure 22.** The hoop strain of the pipeline under the strong vibration mode.

## 4. Conclusions

(1) The increase in burial depth reduces the deformation of the pipeline and the pressure on the pipe top under the action of vibration load, but when the burial depth exceeds 1.8 m, the increase in burial depth contributes little to the reduction of the pressure on the pipe top.

(2) With the increase of compaction times, the backfill soil keeps changing between over-compaction (loose) and compaction states, and the pipe top pressure also keeps changing. Moreover, at a shallow burial, the pipe top pressure is more obvious when the compaction degree changes.

(3) The strain at the top of the pipe is mainly axial strain, while at the side of the pipe, circumferential strain is greater than axial strain. In addition, the strain values at the side of the pipe are all greater than those at the top of the pipe. The axial strain at the pipe top is greatly affected by the adjacent parts, so the trend of its change is different from the pipe pressure with the increase of compaction times.

(4) Under earth pressure, with the increase of buried depth, the soil arching effect at the top of the pipeline becomes more obvious. The calculation results of soil pressure at the top of the pipeline obtained by Chinese specification are quite different from the finite element results, and with an increasing buried depth of the pipeline, the difference between them increases.

(5) Under the self-weight of backfill, with the increase of pipeline burial depth, due to the increase of side soil compaction with the increase of burial depth, the increment rate of lateral deformation of the pipeline is less than that of vertical deformation. At the same time, the increase of pipeline burial depth also increases the compaction of bottom backfill, so the increment rate of pipe top displacement decreases with the increase of buried depth.

(6) Under earth pressure, the elastic modulus of the filling soil has a significant influence on the soil arch effect, and they are positively correlated. However, the increase of elastic modulus of the backfill weakens the attenuation of vibration load in soil. Under the comprehensive effect, the elastic modulus of the backfill within a certain range can effectively reduce the stress deformation of the pipeline under vibration compaction. However, if the elastic modulus of the backfill exceeds 10 times the initial elastic modulus, the deformation of the pipeline will be greater than the initial value.

**Author Contributions:** Writing—original draft preparation, C.M.; writing—review and editing, X.W.; formal analysis, J.Z.; software, H.L.; methodology, Y.Z. All authors have read and agreed to the published version of the manuscript.

**Funding:** This research received no external funding.

**Institutional Review Board Statement:** Not applicable.

**Informed Consent Statement:** Not applicable.

**Data Availability Statement:** The data is unavailable due to privacy restrictions.

**Conflicts of Interest:** The authors declare no conflict of interest.

## Abbreviations

| | |
|---|---|
| HDPE | High-Density Polyethylene |
| VAF | Soil Arch Rate |

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
