# Peer review of "Dynamic Response Analysis of Buried HDPE Pipes under Vibration Compaction Considering the Influence of Buried Depth and Filling Modulus"

_applsci, doi:10.3390/app13063568_

Round 1

Reviewer 1 Report

I  recommend the authors to expand the paper with a discussion of results they obtained. They make an analysis of the research and the results on the pictures.

(Error! Reference source not found) appears in several places in the paper??

Reviewer 2 Report

The dynamic response of buried pipelines under the action of large vibration rollers is studied in this paper experimentally and numerically.

The following comments should be addressed to increase the quality of the paper.

·         All references to Figures and Tables are broken.

·         Manuscript should have had lines numbers to make easier the revision.

·         Section 2.2. Name the regulatory standards that followed the compactness of backfill.

·         Why authors chose those vibration frequencies (33 and 28 Hz)?

·         Equation (2) is moved left.

·         Table 5 is moved left.

·        Many information regarding the numerical model is missing, for instance: which calculation method was used (standard or explicit) , the constitutive model of the pipe, sensitivity analysis with the element size, convergence tolerances if standard method is used, energy balances…

Reviewer 3 Report

The work presented in this paper deals with a very interesting subject, via the results of field tests on the evaluation of the dynamics of buried HDPE pipes under the effects of vibrations. Nevertheless, to improve the presentation quality of this work, I propose some form remarks:

-                     The abstract section is relatively weak, they misses the important points which should be always mentioned in the abstract. Please consider it as an opportunity to attract the attention of potential readers and insert more results,

-                     Please add nomenclature table at the beginning of your manuscript, 

-                     The introduction contains a lot of repetition of the ideas! Please rewrite this part carefully! Please mention relevant literature in the introduction and based on those works highlight the novelty of your paper. Please indicate the limitations of your study,

-                     Please check the text after the automatic insertion of the references, several passages on the text are found ‘Error! Reference source not found’,

-                     The obtained results and there discussions is very limited in length. Please specify how did you validate the performance? This in my opinion totally misses the discussion and interpretation.  Please add more discussion on the obtained results,

-                     The conclusions sections could be improved. Please do not repeat some common and not important phrases. Instead highlight the results of your study and identify future research directions.

Round 2

Reviewer 2 Report

Authos addressed all author comments. The paper is suitable for publication.